# Krüppel-like Factors 4 and 5 in Colorectal Tumorigenesis

**DOI:** 10.3390/cancers15092430

**Published:** 2023-04-24

**Authors:** Esther Lee, Jacky Cheung, Agnieszka B. Bialkowska

**Affiliations:** Department of Medicine, Renaissance School of Medicine, Stony Brook University, Stony Brook, NY 11794, USA

**Keywords:** Krüppel-like factors, intestine, homeostasis, colorectal cancer

## Abstract

**Simple Summary:**

Krüppel-like factors (KLFs) are zinc finger-containing transcription factors that play a crucial role in embryogenesis, development, homeostasis, and disease progression by regulating multiple signaling pathways. This review is focused on the role of two members of the KLF family, KLF4 and KLF5, and their intricate roles in colorectal carcinogenesis.

**Abstract:**

Krüppel-like factors (KLFs) are transcription factors regulating various biological processes such as proliferation, differentiation, migration, invasion, and homeostasis. Importantly, they participate in disease development and progression. KLFs are expressed in multiple tissues, and their role is tissue- and context-dependent. KLF4 and KLF5 are two fascinating members of this family that regulate crucial stages of cellular identity from embryogenesis through differentiation and, finally, during tumorigenesis. They maintain homeostasis of various tissues and regulate inflammation, response to injury, regeneration, and development and progression of multiple cancers such as colorectal, breast, ovarian, pancreatic, lung, and prostate, to name a few. Recent studies broaden our understanding of their function and demonstrate their opposing roles in regulating gene expression, cellular function, and tumorigenesis. This review will focus on the roles KLF4 and KLF5 play in colorectal cancer. Understanding the context-dependent functions of KLF4 and KLF5 and the mechanisms through which they exert their effects will be extremely helpful in developing targeted cancer therapy.

## 1. Introduction

Krüppel-like factors (KLFs) are transcription factors that contain three zinc finger (ZF) domains. Their amino acid sequences are similar to that of the *Drosophila melanogaster* gap gene Krüppel, which plays an essential role during early fruit fly development [1,2,3,4,5]. KLFs have a crucial role in homeostasis, disease development, and progression [1,6,7,8,9,10,11,12,13,14,15,16]. Seventeen KLFs have been identified and studied in multiple disease models so far [17,18]. The ZF domains are highly conserved within the KLF family. Alignment of ZF domains from KLF4 and KLF5 of *Homo sapiens* origin show 82.7% identity [19]. Aside from the three ZF domains, the two KLFs are not structurally similar (Figure 1). Several comprehensive reviews provide analyses of the phylogeny and descriptions of the structure of KLF4 and KLF5 [20,21,22,23,24]. KLF4 and KLF5 undergo multiple post-translational modifications that regulate their transcriptional activity, localization, stability, and degradation. The description below summarizes possible modifications of these two factors based on studies performed in various normal and cancer cell types.

KLF4 protein structure includes an N-terminal activation domain, a repression domain, and a nuclear localization signal (NLS), followed by three ZF domains that interact with DNA (Figure 1). KLF4 transcriptional activity can be induced via methylation of arginine 374, 376, 377 by Protein arginine N-methyltransferase 5 (PRMT5) and sumoylation of lysine 275 through interaction with Small ubiquitin-like modifier 1 (SUMO1) at the SUMO-interacting motif (SIM) located in the N-terminal domain [25,26]. Interestingly, the activation domain of KLF4 interacts with E1A binding protein p300/CREB-binding protein (p300/CBP), leading to the acetylation of the amino acids at position 225 and 229, preventing KLF4 from activating downstream targets [27]. Similarly, the phosphorylation of serine 132 by ERK1/2 reduces transcriptional KLF4 activity and leads to its ubiquitination and degradation [28]. There are several additional mechanisms regulating KLF4 degradation and, thus, reducing its transactivating function. PEST signal, an amino acid sequence rich in proline, glutamic acid, serine, and threonine, located between the activation and repression domains, has been indicated in KLF4′s degradation [29]. Furthermore, lysine residues 32, 52, 232, and 252, located in the activation and repression domains, can be modified by ubiquitination and/or acetylation and participate in KLF4 degradation via the ubiquitin-proteasome pathway [30] (Figure 1).

**Figure 1 cancers-15-02430-f001:**
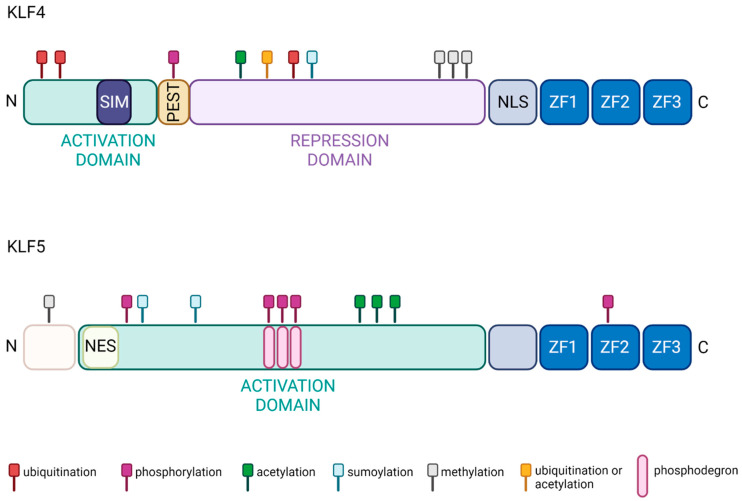
Functional domains of KLF4 and KLF5 proteins. The activity of both proteins is regulated by ubiquitination (red), phosphorylation (purple), acetylation (green), sumoylation (cyan), methylation (graphite), and ubiquitination or acetylation (gold). In addition, three ZF domains are localized at the C-terminus of both proteins and constitute DNA-binding domains [20,21,22,23,24,29,31,32,33,34,35]. Created with BioRender.com (accessed on 12 April 2023).

The KLF5 protein includes a well-defined activation domain that contains a nuclear export signal (NES) followed by three ZF domains in the C-terminal part of the protein (Figure 1). Multiple phosphorylation events regulate KLF5 transcriptional activity. For example, Protein kinase C (PKC) phosphorylates KLF5 at serine 153, which increases its interaction with CBP and the transcriptional activity of this complex [36]. A similar effect on KLF5 activity occurs with acetylation at lysine 369 by acetyltransferase p300 [37,38]. Phosphorylation of the serine residue at position 406 via the Phosphatidylinositol 3′-kinase (PI3K)/-AKT (PI3K/AKT) pathway promotes interaction between KLF5 and Retinoic acid receptor-alpha (RARα) or KLF5 and c-Jun. This interaction leads to increased expression of KLF5′s targets [39,40]. General control non-depressible 5 (GCN5) acetylates KLF5 at lysine 335 and 391 and increases its activity. Notably, KLF5 localization and, thus, activity can be regulated by sumoylation. It has been shown that sumoylation at lysine 151 and 202 leads to inactivation of NES, elevated nuclear levels, and increased activity of KLF5 [41]. In contrast, sumoylation at lysine 162 and 209 by SUMO1 can convert KLF5 from a transcriptional activator to repressor [42]. In addition, the stability of KLF5 is regulated via the ubiquitin/proteasome pathway. There have been three phosphodegron domains identified in the KLF5 protein [43]. For example, Glycogen synthase kinase 3β (GSK3β) and F-box and WD-repeat domain-containing 7 (FBXW7) recognition of the first phosphodegron sequence is essential for phosphorylation of serine at residue 303, leading to KLF5 degradation [35,44,45]. KLF5 methylation by Protein arginine N-methyltransferase 5 (PRMT5) at arginine 57 leads to KLF5 protein stabilization by inhibiting its phosphorylation at the phosphodegron sequence [46] (Figure 1).

Post-translational modifications of KLF4 and KLF5 play an essential role in activity regulation. Thus, we accessed The Cancer Genome Atlas Program (TCGA) and Catalogue of Somatic Mutations in Cancer (COSMIC) databases and acquired a list of mutations in *KLF4* and *KLF5* in samples obtained from CRC patients [47] (Appendix A). Interestingly, there are several mutations in phosphodegron and the regions proximal to it in the KLF5 protein sequences, such as pS303P, pS303L, pS301T, pS301S, pP304L, pP304T, pP304Q, pP304A, pS307A, pS307L, pS311I, and pS311N. In addition, it has been shown that pS303A and pS301S mutations increase KLF5 protein stability and transcriptional activity [35,45]. Thus, it is feasible that mutations near the phosphodegron site affect KLF5 function.

Recent years have brought numerous publications on the role of KLFs in cell proliferation, cell cycle, migration, invasion, and their role in the development and progression of various cancers has been highly appreciated [48,49,50,51,52,53]. KLF4 and KLF5 are studied in-depth in our laboratory, specifically their role in the physiology and pathophysiology of the intestinal epithelium [15]. Initially, KLF4 was identified as a tumor suppressor. However, extensive studies showed that KLF4 could also function as an oncogene. Similarly, KLF5, known and best described as an oncogene under specific circumstances, also has tumor-suppressive functions. KLF4 and KLF5 are involved in the development and progression of multiple cancers, such as breast, lung, gastric, ovarian, pancreatic, prostate, and melanoma, to name a few [22,54,55,56,57,58]. This review will focus on KLF4 and KLF5’s role in colorectal cancer (CRC) tumorigenesis. However, this is not to understate their role in other cancers; instead, research on the development and progression of CRC provides the most comprehensive information.

## 2. Krüppel-like Factor 4

### 2.1. Homeostasis

KLF4 is also called gut-enriched Krüppel-like factor (GKLF). KLF4 regulates transcription by modulating histone H4 acetylation at promoter sites [27]. An overview of its biochemical properties, regulation, and physiological functions was reviewed by us elsewhere [21]. KLF4 plays a significant role during gastrointestinal development and subsequent epithelial homeostasis. During murine fetal development, gastrointestinal KLF4 levels rise on embryonic day 13 and peak at day 17 [59]. By birth, KLF4 levels in colonic cells are typically higher than in small intestine cells. KLF4 levels persist throughout the gastrointestinal tract during life and rise with increasing age throughout adulthood [59]. Specifically, KLF4 is expressed in terminally differentiated epithelial cells at the mucosal villus border and reaches peak levels at terminal differentiation [60,61,62,63]. It is involved in goblet cell differentiation and maintenance and regulation of cell polarity [61]. Conditional ablation of *Klf4* from the intestinal tract resulted in viable mice but with increased rates of epithelial proliferation and migration [64]. Partial depletion of KLF4 in terminally differentiated intestinal cells led to an increase in goblet cells, implying a role for KLF4 in maintaining goblet cell population and mispositioning of Paneth cells, suggesting KLF4-dependent localization [61].

In contrast, deletion of *Klf4* from intestinal epithelial cells early in murine development led to a decrease in the number of colonic goblet cells in adult mice. This is due to differentiation failure, as KLF4 also plays a crucial role in negative regulation of the WNT pathway. Furthermore, Paneth cells were dislocated into the upper crypt due to reduced levels of Ephrin-B1, a KLF4 target [64,65]. Thus, KLF4 plays a vital role in differentiation and maintaining intestinal cell population and organization. Recent studies have shown that KLF4 regulates the proliferation status of a subpopulation of quiescent intestinal stem cells marked by the expression of B Lymphoma Mo-MLV Insertion Region 1 Homolog (*Bmi1*) [66]. Deletion of *Klf4* from *Bmi1*-positive cells during homeostasis led to an increase in *Bmi1*-positive cell proliferation. In contrast, *Bmi1*-specific *Klf4* deletion upon radiation injury reduces levels of Musashi-1 expression and inhibits crypt regeneration, demonstrating the context-dependent function of KLF4 [66,67]. 

Additionally, KLF4 plays an essential role in maintaining genetic stability, initiating apoptosis, and preventing epithelial–mesenchymal (EMT) transition in the progression of CRC. Murine cells that are absent in or have suppressed *Klf4* expression demonstrate higher levels of genetic instability [68,69]. Next, we discuss the contribution KLF4 has made towards the development, prognosis, and treatment of CRC, which will highlight KLF4′s role as both a tumor suppressor and oncogene.

### 2.2. Colorectal Cancer

CRC has the third-highest annual incidence and second-highest mortality among men and women in the United States and worldwide [70]. Every year, an estimated 1.4 million new cases are reported worldwide. Of note, obesity, lack of physical activity, active and passive smoking, and high salt and red meat consumption have been established as risk factors for colorectal cancer [71]. While prevalence and mortality in those aged 50-and-older are declining due to early screening and improving therapies, the everyday nature of risk factors for colon cancer makes it a continued threat. Notably, the incidence of early-onset colorectal cancer in those aged 50 and younger has risen globally between 2.8–36.5% within the last 30 years [72,73,74,75]. Therefore, understanding how KLFs work in the context of colon cancer will be beneficial in preventing and treating colon cancer.

Most CRC development follows a linear framework characterized by the adenoma–carcinoma–metastasis sequence [76]. The most well-studied mutations driving this sequence involve an initial suppression of Adenomatous polyposis coli (APC) followed by overexpression of Kirsten rat sarcoma viral oncogene homolog (KRAS) and loss of Tumor protein 53 (TP53) and Mothers against decapentaplegic homolog 4 (SMAD4) [77]. A review examining the role of KLF4, KLF5, and KLF6 in CRC was published by our group in 2008 [78]. The current review expands on the role of KLF4 and KLF5 in animal models of colorectal cancer, providing recent discoveries in their involvement in regulating the development, progression, and metastasis of CRC. In addition, we described novel pathways that regulate KLF4 and KLF5 activity in CRC and summarized their role in the context of chemotherapy and radiation therapy and their potential as biomarkers of CRC.

KLF4 is decreased in both adenomas from multiple intestinal neoplasia (*Apc^Min/+^*) mice and humans with familial adenomatous polyposis (FAP) when compared to either normal-appearing intestinal tissue from the same individual or healthy controls [79]. KLF4 has been shown to protect against the advancement of colitis into CRC via increased genetic stability in murine models [80]. Immuno-stains of normal colon show a gradient in KLF4 concentration that is the highest near the surface epithelium and lowest towards the crypt [81]. This gradient is disrupted in adenomas and carcinomas [81]. Loss of heterozygosity in *KLF4* and hypermethylation at its 5′-untranslated region are common in CRC [82]. KLF4 primarily contributes to early CRC development and is associated with EMT in CRC [83,84]. The discussion here focuses on KLF4 in solid tumor CRC.

#### 2.2.1. KLF4 Inhibits Cell Cycle Progression and Induces Apoptosis

KLF4 has an established role as a cell cycle regulator. It regulates both the G1/S and G2/M phases of the cell cycle by transcriptional activation of cell cycle inhibitors and inhibition of macromolecular biosynthesis, such as protein and DNA [85]. Overexpression of *KLF4* in colonic adenocarcinoma cells HT-29 reduces cyclin D1 mRNA and protein levels via repression of its promoter at a Specificity Protein 1 (SP1) response element [86]. On the other hand, KLF4 suppression results in HT-29 cells becoming hyperproliferative via increased DNA synthesis activity [81]. Induced *KLF4* overexpression in cells not natively expressing *KLF4*, such as RKO, has also been shown to reduce colony formation via cell cycle inhibition [87]. KLF4 also inhibits centrosome amplification in the presence of DNA damage via induction of p53, followed by a subsequent increase in cyclin E levels and Cyclin-dependent kinase 2 (CDK2) activity [88,89].

Evidence also suggests KLF4 is involved in regulating apoptosis in CRC. KLF4 is a downstream target of interferon-gamma (IFN-γ). Treatment of HT-29 cells with IFN-γ was followed by a time- and dose-dependent increase of *KLF4* transcription and translation independent of p53 [90]. IFN-γ-dependent induction of KLF4 relies on phosphorylated Signal transducer and activator of transcription 1 (STAT1)-mediated activation of the GAS element on the *KLF4* promoter. Interestingly, while *KLF4* overexpression shows evidence of G1/S arrest and apoptosis in HT-29 cells, RKO cells only exhibit G1/S arrest [87]. Studies by our group in RKO cells showed KLF4 was anti-apoptotic, and cells only underwent apoptosis if KLF4 was absent [91]. KLF4 is absent in RKO cells due to promoter hypermethylation, while it is expressed in HT-29 cells.

Peroxisome proliferator-activated gamma (PPAR-γ) activation increases KLF4 transcription and translation [92]. PPAR-γ agonists derived from glycyrrhizin demonstrated cytotoxic activity against CRC via induction of KLF4 [93]. However, it has been noted that PPAR-γ agonists did not consistently induce KLF4, and KLF4 induction can be PPAR-γ independent [94,95]. Separate from PPAR-γ agonists, studies examining histone deacetylase inhibitors (HDACi) show KLF4 does not alter the response to HDACi treatment. Still, apoptosis may instead be modulated by the transcription factors SP1 and SP3 [96]. Overall, KLF4′s role in apoptosis in CRC is context- and cell-dependent. Different responses to the increased KLF4 levels could be due to the endogenous status of KLF4 and mutation status of other genes.

#### 2.2.2. KLF4 Negatively Regulates WNT Pathway Activity

In the context of an *APC*-inactivating mutation, the WNT signaling pathway is hyperactivated in humans and mice [97,98]. In CRC, KLF4 is induced by the transcriptional activation of *APC* [99,100]. This activation depends on Caudal Type Homeobox 2 (CDX2), as mutated *CDX2* has a dominant negative effect on KLF4 [99] (Figure 2). Induction of *KLF4* results in an inverse decrease in the WNT/β-catenin pathway [100,101]. KLF4 binds directly to the transactivation domain on the C-terminus of β-catenin and prevents the recruitment of p300/CBP [65]. Thus, in CRC, the sequence of events appears to be loss of *APC*, resulting in suppression of *KLF4,* followed by increased WNT/β-catenin levels and activity. 

*Apc^Min/+^* is a mouse model carrying a heterozygous nonsense germline mutation of murine *Apc* and serves as an analog for human FAP [102]. *Apc^Min/+^* mice with concomitant whole-body heterozygous deletion of the *Klf4* gene have more intestinal adenomas than *Apc^Min/+^* mice [103]. A follow-up study examining homozygous *Klf4* deletions in the intestinal epithelium of *Apc^Min/+^* mice further demonstrated an increase in the number of intestinal adenomas without an increase in size [104]. 

Complete knockout of intestinal *Klf4* resulted in increased Mechanistic target of rapamycin kinase (mTOR) activity, epigenetic dysregulation, and increased tumorigenicity [104]. Induction of the NOTCH signaling pathway within *Apc^Min/+^* mice also suppressed *Klf4* expression with a concomitant increase in intestinal adenomas and proliferation [105]. In a chemical model of CRC development with Azoxymethane (AOM) as an inducer of DNA mutations, KLF4 has been shown to suppress the frequency of mutations in the *Kras* gene [104]. Taken together, KLF4 might exert its role as a tumor suppressor through various mechanisms by preventing epigenetic modifications via directly blocking protein activity, suppressing mutagenesis, and regulating DNA damage repair.

#### 2.2.3. KLF4 and microRNA in CRC

In general, microRNAs (miRs) play a crucial function in gene regulation and cellular homeostasis. Given the effect of miRs on *KLF4* expression (Table 1), specific miR inhibitors serve as potential therapeutic targets against CRC progression. Conversely, KLF4 has also been shown to regulate the expression of multiple miRs. For example, KLF4 positively regulates the expression of miR-153-1 by directly binding to its promoter, and *KLF4* overexpression results in increased miR-153-1 levels and reduced CRC cell proliferation and migration [106]. miR-144, which has been shown to increase stemness in CRC, regulates KLF4 and contributes to the maintenance of cancer stem cells in CRC by increasing their proliferation and invasion [107].

#### 2.2.4. KLF4 Regulates Stemness of CRC

KLF4 is one of the Yamanaka factors, alongside Octamer-binding protein 3/4 (OCT3/4), SRY-box transcription factor 2 (SOX2), and Cellular myelocytomatosis oncogene (c-MYC). Yamanaka factors function as reprogramming factors whose expression is critical for embryonic development and induce a pluripotent state in differentiated cells, thereby driving CRC stem cell activity [125,126,127]. Recent studies have demonstrated increased KLF4 levels in cancer stem cell (CSC) spheroids generated from CRC cell lines [128]. In the spheroids, KLF4 expression at mRNA and protein levels was increased compared to adherent cells. Forced expression of *KLF4*, *OCT3/4*, *SOX2*, and *c-MYC* in CRC cell lines increased CSC properties of a sub-population of CRC stem cells. These properties include increased resistance to chemotherapeutics, spheroid formation, and tumorigenicity in vivo [129,130]. Similarly, overexpression of *KLF4* in Lgr5^+^CD44^+^EpCAM^+^ CSCs resulted in activation of the Transforming Growth Factor Beta-1 (TGFβ-1) pathway, increased stemness, and mesenchymal phenotype [131]. Increased levels of KLF4 and other stemness markers were observed in the CD133+ CRC stem cell population, which is responsive to the WNT/β-catenin pathway which promotes tumor growth and metastasis. Notably, even CD133- metastatic cells have elevated levels of KLF4 [132]. The same study showed an increase in embryonic stem cell signatures within the tumors of metastatic patients, including markers such as KLF4 and BMI1 [132]. Singovski and colleagues suggest that metastatic reprogramming could result from the activation of the Hedgehog/GLI pathway and repression of the WNT/β-catenin pathway, leading to the activation of pluripotent gene expression [130]. These exciting findings show a KLF4 context-dependent role in CRC and might provide new targets for CRC treatment.

#### 2.2.5. KLF4 as a Therapeutic Target in CRC

Given its role as a tumor suppressor, induction of KLF4 represents a potential pathway to CRC treatment. NOTCH inhibitors, such as dibenzoazepine, and gamma-secretase inhibitors demonstrate anti-tumor activity and have been shown to induce expression of *KLF4* [105,133]. ML-133 is a small molecule that increases KLF4 levels by reducing intracellular levels of labile zinc [134]. As a result, KLF4 replaces SP1 on the *CCND1* promoter and inhibits Cyclin D1 expression and CRC proliferation. PPAR-γ agonists have been shown to induce KLF4 levels and inhibit CRC growth [92,93,94]. Acetylbritannilactone, an isolate from a traditional Chinese medicinal (TCM) herb, induces cell cycle arrest via a KLF4-dependent stimulation of p21 expression [135]. Sijunzi decoction, another TCM substance prescribed for CRC, has been shown to upregulate KLF4 [136]. However, as mentioned above, KLF4 might also act as an oncogene in CRC by regulating CSC proliferation and metastasis. Thus, a comprehensive evaluation of therapeutics targeting KLF4 in different CRC cell populations is paramount.

The presence, or lack thereof, of KLF4 also modulates the response of CRC cells to chemotherapeutics. It has been shown that KLF4 promotes HCT-15 sensitivity to cisplatin cytotoxicity and cisplatin-mediated G2/M cell cycle arrest [137]. While KLF4 typically negatively regulates High mobility group box 1 (HMGB1) and human Telomerase reverse transcriptase (hTERT), the reverse was confirmed in the presence of cisplatin [137]. KLF4 was also found to be downregulated in oxaliplatin-resistant CRC cells. The KLF4 mutation p.A472D may also confer resistance to cetuximab in CRC [138]. 

KLF4 may also modulate radiotherapy response. Mex-3 RNA binding family member A (MEX3A) has been shown to enhance radio-resistance. KLF4 and its agonist suppress *MEX3A*, thereby potentially increasing radio-sensitivity [139]. Radio-resistant CRC has also been shown to have elevated levels of RAP1 GTPase-activating protein (RAP1), which has a strong physical interaction with KLF4, suggesting modulation of radio-sensitivity via the RAP1/KLF4 axis [140]. Interestingly, overexpression of BMI1, which is positively correlated with KLF4 levels in CRC, has also been linked to radio-resistance [141]. The overall effect of KLF4 on radiotherapy is not apparent and should be studied in depth, as it is associated with increased and decreased responses to radio-sensitivity.

#### 2.2.6. KLF4 as a Biomarker of CRC

There is conflicting evidence regarding the prognostic value of KLF4 expression in CRC. Our initial data suggested KLF4 is not a marker of tumor stage, size, location, differentiation, or metastasis [84]. However, loss of KLF4 is an independent predictor of survival, recurrence, and poor differentiation [142,143,144]. A KLF4 activity score has been proposed and associated with CRC infiltration into myeloid lineage cells [145]. Interestingly, higher levels of KLF4 in normal tissues of CRC patients were shown to be associated with higher recurrence and lower overall survival [146]. KLF4 has also been shown to be increased in normal human epithelial cell NCM460 and colorectal epithelial cancer cell Caco-2 and HCT116, which can promote migration of cells in NCM460 and HCT116 [147]. KLF4 also negatively regulates GINS4, which is overexpressed in CRC and correlated with advanced staging and differentiation [148], suggesting that KLF4 plays at least an intermediary role in prognostication. However, given its cell- and context-dependent behaviors, KLF4 cannot currently be used reliably as a biomarker.

## 3. Krüppel-like Factor 5

### 3.1. Homeostasis

KLF5 is also known as Intestinal Krüppel-like factor (IKLF) due to its high expression in intestinal epithelium. However, KLF5 can be detected in almost all tissues, including breast, prostate, pancreas, intestine, lung, bladder, and skeletal muscle [22,32,60,149,150,151]. KLF5 regulates many cellular processes, including cell cycle, proliferation, migration, invasion, stemness, apoptosis, and autophagy, and plays a crucial role in maintaining gut homeostasis [32,152,153,154,155,156,157,158,159,160,161,162,163]. Notably, KLF5 regulates villus formation and initiates cytodifferentiation in embryonic intestinal epithelium. Deletion of *Klf5* from intestinal epithelium during embryogenesis leads to downregulation of multiple genes such as E74-like ETS transcription factor 3 (*Elf3*), *Pparg*, Atonal BHLH transcription factor 1 (*Atoh1*), Achaete-scute family bHLH transcription factor 2 (*Ascl2*), Hepatocyte nuclear factor 4 alpha (*Hnf4a*), Neurogenin 3 (*Neurog3*), and Caudal Type Homeobox 1 (*Cdx1*) [164]. 

Similarly, data obtained from *Klf5* deletion in the gut suggest that KLF5 plays a role in maintaining epithelial proliferation, differentiation, and cell positioning along the crypt radial axis in adult mice [165,166]. Mice with deletion of *Klf5* within active intestinal epithelial stem cells have decreased expression of intestinal stem cell signature genes, such as *Lgr5*, Olfactomedin 4 (*Olfm4*), and *Ascl2*, and impaired stem cell renewal. KLF5 is crucial for stem cell activity and regeneration of the intestinal epithelium after injury [167,168]. KLF5 also regulates DNA damage repair in intestinal epithelial cells upon radiation injury. In mice with heterozygous deletion of *Klf5* in intestinal epithelial cells, genes involved in nucleotide excision repair, mismatch repair, and non-homologous end-joining were significantly downregulated compared to wild-type mice [169]. Mice with intestinal epithelium-specific deletion of *Klf5* also developed a Th-17-mediated immune response and subsequent colitis, suggesting a protective role of KLF5 against intestinal inflammation [170]. 

Evidently, KLF5 is indicated in a wide range of processes to ensure intestinal epithelial homeostasis in the presence of insults. While the lack of KLF5 activity can lead to insufficient self-renewal and intestinal integrity, overactivation of KLF5 may cause uncontrolled cell proliferation and differentiation, ultimately leading to tumorigenesis. As such, understanding the role of KLF5 in achieving balance in these cellular processes is essential to ensure intestinal health. However, whether KLF5 functions to upregulate or downregulate these processes is context-dependent and highly controversial.

### 3.2. Colorectal Cancer

#### 3.2.1. KLF5 Is a Pro-Proliferative Factor in CRC

KLF5 is a pro-proliferative transcription factor downstream of the classical Mitogen-activated protein kinase (MAPK-ERK-RAS) pathway and directly regulated by Early Growth Response 1 (EGR1) [171]. Activation of the KRAS oncogene plays an essential role in CRC pathophysiology, and KLF5 contributes to colorectal tumorigenesis induced by a constitutively activating KRAS mutation (G12V) (Figure 3). For example, Klf5 haploinsufficiency in Apc^Min/+^/Kras^G12V^ mice resulted in significantly reduced tumor number and size compared to Apc^Min/+^ mice [172]. In addition, increased levels of KLF5 were observed in spontaneous hyperplastic intestinal polyp development and colonic tumorigenesis in Villin-Cre/LSL-KRAS^G12D^ mice, further supporting KLF5′s role as a mediator of the KRAS pathway in CRC formation [173]. Interestingly, while the Villin-Cre/LSL-KRAS^G12D^ mice displayed decreased survival when treated with AOM compared to controls, loss of one Klf5 allele showed reduced levels of KRAS effector proteins and, as a result, reduced mortality upon AOM treatment [173]. Overall, KLF5 expression appears essential in exerting the oncogenic, pro-proliferative effects of KRAS mutations in CRC. 

The HIPPO pathway regulates cell stemness and proliferation via two key transcriptional coactivators, Yes1 Associated Transcriptional Regulator (YAP1) and WW domain-containing transcription regulator protein 1 (TAZ). The KLF5-YAP1 complex induces transcription of *Ascl2*, a WNT signaling target, to ensure the self-renewability of CRC progenitor cells [158]. Synaptopodin-2 (SYNPO2) was shown to inhibit the KLF5-YAP signaling pathway and suppress hypoxia-induced progression of CRC [174].

The TGF-β/SMAD4-signaling pathway and its role in CRC are well-established. Silencing *KLF5* was found to sensitize SMAD4-deficient cells to TGF-β-induced apoptosis. Conversely, overexpression of *KLF5* significantly inhibited TGF-β-induced apoptosis in SMAD4-proficient cells. This suggests that KLF5 acts as an oncogene in CRC regardless of SMAD4 expression [175]. One recent study discovered that primary mesenchymal stromal cells (MSCs) play a dual role in regulating C-X-C Motif Chemokine Ligand 5 (CXCL5), which is significantly overexpressed in CRC, allowing for distant metastasis and angiogenesis. MSCs not only secreted C-C Motif Chemokine Ligand 7 (CCL7) to promote acetylation of KLF5 and upregulate transcription of CXCL5 but also secreted TGF-β to regulate SMAD4 and reverse the effect of KLF5 on the transcription of CXCL5 [176].

#### 3.2.2. KLF5-WNT/β-Catenin Positive Feedback Loop Regulated CRC Development and Progression

Germline loss-of-function mutation of *APC* and mutations in *CTNNB1* have been identified as a cause of colorectal cancer. APC plays an essential role in regulating the activity of β-catenin, which controls the WNT signaling pathway responsible for maintaining the proliferation of the intestinal crypt epithelium. KLF5 is a crucial mediator of these interactions contributing to CRC tumorigenesis. *Klf5* haploinsufficiency in the context of *Apc* mutation was associated with lower levels and reduced nuclear localization of β-catenin, resulting in reduced expression of *Ccnd1* and *c-Myc*, downregulation of the WNT pathway activity, and decreased polyp formation [177]. In addition, the formation of lethal colorectal adenomas and carcinomas induced by β-catenin mutations in Lgr5^+^ stem cells was entirely suppressed by *Klf5* deletion [178]. Overall, lack of KLF5 expression prevented the tumorigenic effects of *Apc* mutation and β-catenin activation, suggesting the oncogenic function and necessity of KLF5 in CRC (Figure 3).

Lysophosphatidic acid (LPA), a simple phospholipid with potent mitogenic effects, and its receptor LPAR modulate the tumorigenic effects of *APC* mutation. Compared to *Apc^Min/+^* mice, *Apc^Min/+^/Lpar2*^−/−^ mice exhibited decreased tumor progression and hypoxia in response to reduced expression of *Klf5*, *Ctnnb1*, *Ccnd1* and *c-Myc* [179]. A recent study proposed a new mechanism by which KLF5 modulates the WNT/β-catenin pathway in the presence of LPA. Contrary to previous findings, silencing KLF5 did not alter the nuclear translocation of β-catenin by LPA. Instead, KLF5 was found to facilitate LPA-induced formation and transcriptional activity of the β-catenin/TCF complex to promote colon cancer cell proliferation [180].

Ketogenesis is significantly decreased in the tumor microenvironment of CRC. As such, a ketogenic diet of high lipids and low carbohydrates has been recommended for cancer patients. Increasing ketogenesis markedly decreased KLF5-dependent synthesis of C-X-C Motif Chemokine Ligand 12 (CXCL12) in cancer-associated fibroblasts, ultimately increasing the infiltration of immune effector cells in tumors and enhancing sensitivity to immune checkpoint inhibitors specific for programmed cell death 1 (PD-1) [181]. By the same mechanism, increasing ketogenesis inhibited CRC migration, invasion, and metastasis both in vitro and in vivo [182].

#### 3.2.3. KLF5 and microRNA in CRC

MiRs bind directly to the 3′UTR of *KLF5*, thereby suppressing colorectal cancer cell proliferation, migration, and stemness in vitro and inhibiting tumor growth in vivo in mouse models. Recent studies have found several miRs that target and modulate KLF5 at the post-transcriptional level to regulate the development of CRC (Table 2).

MiR-143 and miR-145 have been found to decrease the expression of *KLF5* in CRC [183]. Consistent with this finding, one study suggests that increased expression of the long intergenic noncoding RNA (lncRNA) LINC00908 may act as a competing endogenous RNA to negatively regulate the miR-143-3p/KLF5 axis, thereby promoting cell proliferation and survival of colorectal cancer cells [185]. miR-4711-5p was also shown to bind directly to the 3′UTR of *KLF5*, thereby suppressing colorectal cancer cell proliferation, migration, and stemness in vitro and inhibiting tumor growth in vivo in mouse models [186]. Overexpression of miR-143-3p was also associated with downregulation of *KLF5* and was detected in significantly lower amounts in more advanced CRC [187]. 

In addition, lncRNAs have been identified as targets of KLF5 in CRC. For example, lncRNA plasmacytoma variant translocation 1 (PVT1) was found to be regulated by its upstream transcription factor KLF5 and was detected in significant amounts in CRC [188]. Small Nucleolar RNA Host Gene 12 (SNHG12) was also proposed as a lncRNA target for KLF5, positively regulating CRC invasion and distal metastasis. However, whether targeting KLF5-SNHG12 will produce therapeutic benefits is still being investigated [189]. Another study proposes a novel mechanism by which the KLF5 protein constructs a loop-like three-dimensional genome structure consisting of *KLF5* promoter, enhancer, and the transcription start site region of Colon Cancer Associated Transcript 1 (CCAT1). This promoter-enhancer loop may modulate the expression of KLF5 and CCAT1, resulting in the maintenance of colorectal cancer stemness [157]. Recently, low-molecular-weight compounds targeting the hydrophobic α-helix structure of KLF5, known as a potential interface for protein–protein interaction, were synthesized using pyrazinooxadiazine-4,7-dione. Once bound to this interface, these compounds selectively suppressed levels of the KLF5 protein and reduced the expression of proteins involved in the WNT signaling pathway, thereby inhibiting the proliferation and survival of transplanted colorectal cancer cells in vivo [190].

#### 3.2.4. KLF5 as a Therapeutic Target in CRC

Using an ultra-high-throughput screen, our group identified two KLF5-selective compounds, CID 439501 and 5951923, that significantly decrease endogenous KLF5 protein levels and reduce the viability of several CRC cell lines [191]. A small-molecule compound called ML264 was found to be a KLF5 inhibitor, preventing the expression of *KLF5* and the growth of CRC xenograft tumors [192]. ML264 exerted this effect by inhibiting the RAS/MAPK/PI3K and the WNT/β-catenin signaling pathway. The same KLF5 inhibitor was recently used to investigate CRC resistance to oxaliplatin, a first-line chemotherapy drug commonly used in CRC. Using ML264, the study successfully inhibited the KLF5/BCL-2/Caspase 3 signaling pathway, thereby restoring the apoptotic response and significantly restoring sensitivity to oxaliplatin in CRC patient-derived organoids [193]. Interestingly, SR18662, a derivative of ML264, demonstrated enhanced abilities to inhibit KLF5, the MAPK and WNT pathways, and the growth of CRC in vitro and in vivo with the ability to exert cytotoxic effects [194]. Dual-specificity phosphatase 10 (DUSP10), known for its role in deactivating MAP kinases, reduced intestinal epithelial cell proliferation via inhibition of ERK1/2 activation and KLF5 expression [195]. 

KLF5 protein stability can also regulate CRC. The pP301S KLF5 mutation blocks the recognition of its phosphodegron sequence by F-box and WD-repeat domain-containing 7α (FBXW7α), an interaction essential for the proper degradation of KLF5. As a result of this mutation, KLF5 becomes more stable and resistant to GSK3β-dependent degradation, enhancing its oncogenic abilities in CRC [35]. Heterozygous FBXW7 propellor tip (pR482Q) mutations have also been found to elevate KLF5 levels and promote intestinal tumorigenesis in the presence of the *Apc* mutation [196]. Frequent missense mutations were also found within the second phosphodegron domains of KLF5, which increased KLF5 protein stability by preventing KLF5 and FBXW7 interaction and ultimately led to increased cell proliferation in CRC [43].

Recently, anti-EGFR therapy has been gaining attention as a potential clinical option for CRC. EGFR combined with prion protein was found to activate both KLF5 and Forkhead Box O3 (FOXO3a), a downstream effector of EGFR and the PI3K pathway, thereby inducing cisplatin resistance in aggressive CRC. Therefore, this prion-FOXO3a-KLF5 axis may be a helpful predictor of chemotherapy resistance and patient outcome [197]. However, some data show that elevated expression of KLF5 is associated with increased tumor-infiltrating immune cells and suggest that upregulation of KLF5 may be linked to a better prognosis in CRC [198].

KLF5 also modulates CRC response to radiation therapy. For example, HCT116 cells with significantly higher levels of KLF5 were shown to increase CyclinD1 and β-catenin and promote better cell viability than control cells when subjected to radiation therapy [199]. In addition, the depletion of KLF5 in HCT116 cells increased CRC sensitivity to DNA-damaging ultraviolet irradiation therapy by failing to induce the proto-oncogene, serine/threonine kinase 1 (PIM1) survival kinase [200]. It appears that overexpression of *KLF5* confers resistance to radiotherapy, while reduction of KLF5 may increase susceptibility to radiotherapeutic effects in CRC.

#### 3.2.5. KLF5 as a Biomarker of CRC

Overexpression of *KLF5* may be used as a predictive biomarker for poor tumor regression after preoperative chemoradiation therapy, the standard treatment for locally advanced rectal cancer [199]. A recent study was the first to examine the expression of levels of KLF5 in patients with colorectal cancer to determine correlation with clinical outcomes. The study revealed that high expression of KLF5 in tissues collected from CRC patients was associated with vascular invasion, increased serum carbohydrate 19-9, larger metastatic liver tumors, and poorer prognosis after surgery. While further investigation is needed, KLF5 upregulation of *c-MYC* and *CCND1* via promoter binding may be the mechanism underlining these effects. Thus, high KLF5 expression can independently predict poor prognosis in patients with primary CRC and liver metastasis [201]. However, KLF5 and its use as a prognosis marker in CRC must be studied further.

## 4. Conclusions

The last decade has brought much information regarding the roles KLF4 and KLF5 play in cancer development and progression. Notably, much has been learned about the context-dependent functions of both KLFs and their function during homeostasis, cancer development, and advancement. In addition, both factors can act as a tumor suppressor or oncogene. Thus, an in-depth understanding of mechanisms regulating their expression and activity on epigenetic, transcriptional, translational, and post-translational levels will help to interpret KLFs’ roles in CRC progression. Furthermore, considering the significance of KLF4 and KLF5 in tumor formation and development, efforts to understand their context-dependent role and downstream effectors, interacting proteins, and response to chemotherapy drugs are essential.

Targeting a transcription factor with a small molecule modulator is difficult. There is no available protein structure of KLF4 and KLF5, except well-characterized ZF domains that are highly conserved between SP and KLF family members. Thus, certain limitations exist in designing specific inhibitors or activators for these factors. Therefore, identifying the target genes, lncRNAs, and miRs that regulate the expression of KLF4 and KLF5 is paramount to fully understanding their function while developing targeted therapies. Another obstacle in utilizing KLF4 and KLF5 as therapy targets is their context-dependent roles. For example, KLF4 is a tumor suppressor; however, it becomes indispensable as a proliferative factor during intestinal epithelium regeneration following radiation injury. As such, manipulating these factors’ levels and activity and the treatment’s tentative timing needs to be well thought out. In this review, we have only focused on two out of seventeen KLF family members and narrowed the scope to a single cancer type: colorectal. Considering that KLFs have overlapping roles in various tissues, it is imperative to expand our understanding of the role any/all KLFs play in cancers of all tissue types.

## Figures and Tables

**Figure 2 cancers-15-02430-f002:**
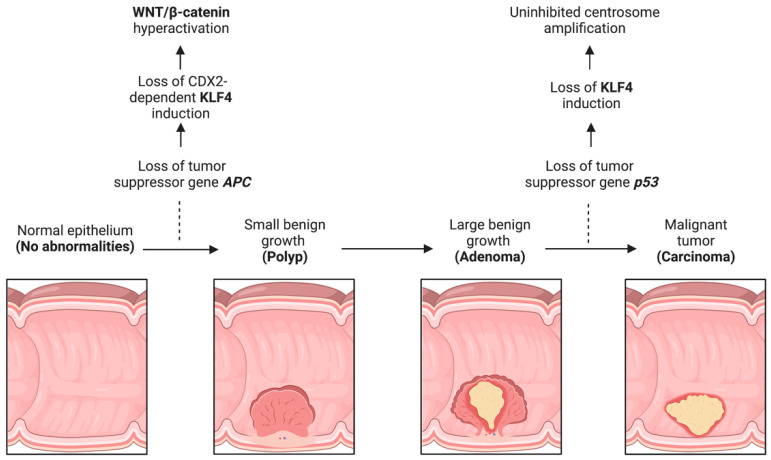
KLF4 function during CRC progression. CRC progression is prototypically defined by the adenoma–carcinoma–metastasis sequence, which typically involves loss of *APC* followed by loss of *p53*. Loss of these genes leads to KLF4 dysregulation with subsequent hyperactivation of WNT/β-catenin and centrosome amplification. Created with BioRender.com (accessed on 12 April 2023).

**Figure 3 cancers-15-02430-f003:**
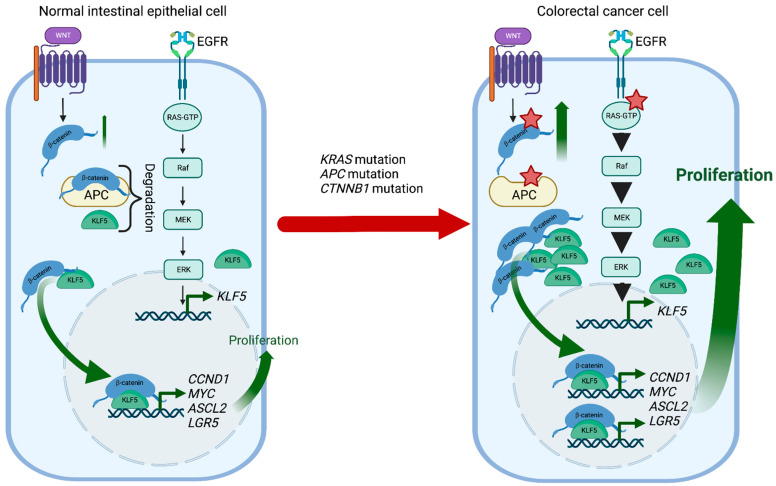
KLF5 and WNT signaling in CRC development. In normal intestinal epithelial cells, KLF5 and β-catenin are regulated by multiple mechanisms such as degradation or well-coordinated WNT and MAPK kinase activation. During CRC development, *Kras* mutations increase *KLF5* expression, while *Apc* and *Ctnnb1* mutations increase WNT pathway activity by increasing the stability and transcriptional activity of β-catenin. In the context of these mutations, KLF5 and β-catenin contribute to CRC tumorigenesis by inducing transcription of multiple genes such as *Ccnd1*, *c-Myc*, *Ascl2*, or *Lgr5*. Red stars mark mutations. Created with BioRender.com (accessed on 12 April 2023).

**Table 1 cancers-15-02430-t001:** MiRs regulating *KLF4* in CRC.

MiRs	Effect on *KLF4*	Mechanism/Biological Data	Reference
miR-7-5p	↓	Inhibits CRC proliferation and migration by suppressing KLF4	[108]
miR-10a	↓	KLF4 is upregulated in *miR-10a* knockout mice	[109]
miR-10b	↓	Directly targets *KLF4* and has been shown to be upregulated in metastatic CRC	[110]
miR-25-3p	↓	Targets *KLF4* with downstream consequences on vascular permeability and angiogenesis	[111]
miR-34a	↓	Targets *KLF4*, function undetermined	[112]
miR-92a	↓	Directly targets and downregulates *KLF4* and *Cdkn1a*, thereby allowing for cell proliferation via upregulation of WNT/β-catenin pathway. Expression of miR-92a also increases CRC metastasis through *KLF4* targeting and subsequent stimulation of MMP2 and inhibition of E-cadherin	[113,114,115,116,117,118,119]
miR-103	↓	Associated with metastatic potential and targets *KLF4*	[120]
miR-107	↓	Associated with metastatic potential and targets *KLF4*	[120]
miR-152-3p	↓	Inhibits *KLF4* expression and is overexpressed in CRC cells	[121]
miR-205	↑	Causes expansion of mucus-secreting goblet-like cells with associated induction of KLF4	[122]
miR-375	↓	Binds to 3′UTR of *KLF4* and inhibits *KLF4*	[123]
miR-543	↓	Promotes CRC proliferation and metastasis via direct targeting of *KLF4*	[124]

Note. ↓—downregulation, ↑—upregulation.

**Table 2 cancers-15-02430-t002:** MiRs regulating *KLF5* in CRC.

MiRs.	Effect on *KLF5*	Mechanism/Biological Data	Reference
miR-143	↓	Works with miR-145 to directly target and inhibit *KLF5*, which is required for KRAS-mediated transformation of normal colonic epithelium	[183]
miR-145	↓	Works with miR-143 to directly target and inhibit *KLF5*, which is required for KRAS-mediated transformation of the normal colonic epithelium. It also directly targets *KLF5* and induces cell cycle arrest in the G1 phase	[183,184]
miR143-3p	↓	Directly targets and suppresses *KLF5*, thereby regulating the cell cycle and promoting intrinsic apoptosis of CRC	[185]
miR-4711-5p	↓	Directly binds to the 3′UTR of *KLF5* and suppresses CRC proliferation, migration, and stemness	[186]

Note. ↓—downregulation.

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
