# Peer review of "Krüppel-like Factors 4 and 5 in Colorectal Tumorigenesis"

_cancers, 2023, doi:10.3390/cancers15092430_

Round 1

Reviewer 1 Report (Previous Reviewer 1)

My concerns have been addressed

Author Response

We want to thank the Reviewer for their kind response.

Reviewer 2 Report (Previous Reviewer 2)

My suggestions:

1. For Figure 1, I would also add the mutations in KLF4 and KLF5 to the figure to see in which domain they were located. Or I would add a table in the manuscript, which reflects the possible effect of mutations, and on which domain they are located.

2. Are KLF4 and  KLF5 involved in another kind of cancer or non-cancer-related diseases?

Author Response

Please see our response below:

Comment 1. For Figure 1, I would also add the mutations in KLF4 and KLF5 to the figure to see in which domain they were located. Or I would add a table in the manuscript, which reflects the possible effect of mutations, and on which domain they are located.

Response 1. We have two Supplementary Tables (Table 1 for KLF4 and Table 2 for KLF5) listing mutations/deletions of these genes that might affect their function. In each table, we added a column called "Location - domain" and listed domains where these modifications occur. Unfortunately, we could not include it in Figure 1, as there are too many mutations/deletions, and the figure would be difficult to read.

Comment 2. Are KLF4 and KLF5 involved in other kinds of cancer or non-cancer-related diseases?

Response 2. Yes, both KLFs play essential roles in maintaining the homeostasis of multiple tissues, regulating inflammation and regeneration, and participate in the carcinogenesis of various cancers. We added one statement in the abstract regarding these roles. One statement in the introduction already addresses their functions in different cancers. The Editor of the special issue asked us to narrow the scope of this review, and as such, we did not discuss other diseases or cancers.

This manuscript is a resubmission of an earlier submission. The following is a list of the peer review reports and author responses from that submission.

Round 1

Reviewer 1 Report

Lee et al. provide a review on some of the major roles of KLF4 and KLF5 in normal intestinal cells and colorectal cancer. Switching to solely CRC has made it much easier to read. There are a few minor areas that could be modified that can enhance and clarify this review. 

-In the beginning, when discussing CRC and how levels have been declining, it may be useful to mention despite this that levels of CRC are increasing in younger populations.  Has anyone looked at KLF4 or KLF5 in younger patients to see if there is a difference?

-It’s mentioned that the group published a review on KLF4, KLF5, and KLF6 in colon cancer in 2008. How does this one differ? Are you only focusing on new evidence or expanding on what was published previously? If you are mentioning the previous review, then you should explain how this differs in the text.

-For KLF4, loss seems to be the major effect in CRC, but there is one sentence that suggests KLF4 is important for cancer stem cells. This should be explained more regarding this contradiction.

-Lines 107-113 seem to be largely repeats from earlier in the text and it’s unclear why talking about normal development here in the CRC section

-When talking about interferon gamma, the notation should be IFN-Y not INF-Y

-Lines 143-145 it is unclear why HDACis are being discussed. Is this in the presence of PPAR-y agonists?  That is unclear in the text.   

-For figures 2 and 3, more information should be included in the figure legend besides see text.

-Does KLF5 regulate miRNAs? Is KLF5 regulated by miRNAs?

-Since KLF4 and KLF5 have such opposite effects in colon cancer, it might be beneficial to have a summarizing image or table to show the similarities/differences between the two

Author Response

March 12th, 2023

Editor and Reviewers

Cancers

Dear Editor and Reviewers,

We want to thank you for your comments and suggestions. This revised manuscript addressed all the comments and suggestions the Editor and Reviewers provided. Please see our responses to your comments below.

Reviewer 1.

Comment 1. In the beginning, when discussing CRC and how levels have been declining, it may be useful to mention despite this that levels of CRC are increasing in younger populations. Has anyone looked at KLF4 or KLF5 in younger patients to see if there is a difference.

Response 1. We added a statement regarding the rise of the early onset of colorectal cancer and provided references. Unfortunately, no one has determined any differences in the KLF4 or KLF5 levels in the younger population.

Comment 2. It’s mentioned that the group published a review on KLF4, KLF5, and KLF6 in colon cancer in 2008. How does this one differ? Are you only focusing on new evidence or expanding on what was published previously? If you are mentioning the previous review, then you should explain how this differs in the text.

Response 2. We provided a short statement to understand the difference between both reviews better.

Comment 3. For KLF4, loss seems to be the major effect in CRC, but there is one sentence that suggests KLF4 is important for cancer stem cells. This should be explained more regarding this contradiction.

Response 3. We included a paragraph describing KLF4's role as a stemness marker and discussed its potential mode of regulation in CRC stem cells.

Comment 4. Lines 107-113 seem to be largely repeats from earlier in the text and it’s unclear why talking about normal development here in the CRC section.

Response 4. We moved parts of this section to the section related to the KLF4 role in the normal intestinal epithelium and one sentence to the next paragraph. We updated the citations accordingly.

Comment 5. When talking about interferon gamma, the notation should be IFN-Y not INF-Y.

Response 5. We made appropriate modifications.

Comment 6. Lines 143-145 it is unclear why HDACis are being discussed. Is this in the presence of PPAR-y agonists? That is unclear in the text.

Response 6. We modified the text to improve its clarity.

Comment 7. For figures 2 and 3, more information should be included in the figure legend besides see text.

Response 7. We provided more information in both figures’ legends.

Comment 8. Does KLF5 regulate miRNAs? Is KLF5 regulated by miRNAs?

Response 8. We updated the section regarding miRs and KLF5 and added table 2, listing miRs regulating KLF5 expression.

Comment 9. Since KLF4 and KLF5 have such opposite effects in colon cancer, it might be beneficial to have a summarizing image or table to show the similarities/differences between the two.

Response 9. We provided an example of the opposite effects of KLF4 and KLF5 on the WNT signaling pathway in Figures 2 and 3 in the original manuscript. KLF4 is shown to work as a tumor suppressor, while KLF5 is an oncogene. KLF4 and KLF5 have been studied in the context of different pathways, and thus, it is rather challenging to provide a concise regulatory network.

Reviewer 2.

Comment 1. Besides colorectal cancers, what kind of cancers KLF4 and KLF5 could be involved in? Authors may briefly mention in in the introduction.

Response 1. In the introduction section, we listed several cancers involving both transcription factors and provided additional citations.

Comment 2. It would be nice to introduce KLF4 and KLF5 structures in the text too in more detail. I would also mention the significance of specific sites for ubiquitination, phosphorylation, acetylation, and sumoylation.

Response 2. We added a section in the introduction that addresses the posttranslational regulation of KLF4 and KLF5. In addition, we added two figures with protein alignment for KLF4 and KLF5 from Mus musculus and Homo sapiens and marked modified residues on these figures.

Comment 3. Were there any genetic mutations in KLF4 and KLF5 reported, involved in cancer, or in any diseases? If yes, which mutations were reported to impact cancer (or another disease?). A table may be needed for it.

Response 3. We accessed TCGA and COSMIC databases and identified mutations in KLF4 and KLF5 in CRC patients. These mutations are listed in Supplementary Tables 1 and 2. In addition, we added a short paragraph addressing some of the identified mutations.

Comment 4. A table may be needed on miRNAs, which may impact KLF5 regulation.

Response 4. We included Table 2, listing miRs regulating KLF5 expression.

Comment 5. Could the KLF4 and KLF5 mutations/dysfunction impact the radiotherapy response of patients?

Response 5. We provided information regarding the role of KLF4 and KLF5 in response to radiotherapy.

We revised the manuscript and addressed all the comments. We hope the incorporated changes will satisfy the Reviewers and Editor and render the revised manuscript suitable for publication. Thank you for being so considerate.

We confirm that neither the manuscript nor any parts of its content are currently under consideration or published in another journal.

All authors have approved the manuscript and agree with its submission to the Cancers.

Please feel free to contact me if I can be of further assistance.

Sincerely Yours,

Agnieszka B. Bialkowska, PhD

Associate Professor

Renaissance School of Medicine at Stony Brook University

Department of Medicine

GI Translational Research Lab

HSC-T17 Room 090

Stony Brook, NY 11794-8176

Phone: (631) 638 2161

Email: Agnieszka.Bialkowska@stonybrookmedicine.edu

Reviewer 2 Report

My suggestions:

1. Besides colorectal cancers, what kind of cancers KLF4 and KLF5 could be involved in? Authors may briefly mention in in the introduction. 

2. It would be nice to introduce KLF4 and KLF5 structures in the text too in more detail. I would also mention the significance of specific sites for ubiquitination, phosphorylation, acetylation, and sumoylation. 

3. Were there any genetic mutations in KLF4 and KLF5 reported, involved in cancer, or in any diseases? If yes, which mutations were reported to impact cancer (or another disease?). A table may be needed for it.

4. A table may be needed on miRNAs, which may impact KLF5 regulation.

5. Could the KLF4 and KLF5 mutations/dysfunction impact the radiotherapy response of patients? 

Author Response

March 12th, 2023

Editor and Reviewers

Cancers

Dear Editor and Reviewers,

We want to thank you for your comments and suggestions. This revised manuscript addressed all the comments and suggestions the Editor and Reviewers provided. Please see our responses to your comments below.

Reviewer 1.

Comment 1. In the beginning, when discussing CRC and how levels have been declining, it may be useful to mention despite this that levels of CRC are increasing in younger populations. Has anyone looked at KLF4 or KLF5 in younger patients to see if there is a difference.

Response 1. We added a statement regarding the rise of the early onset of colorectal cancer and provided references. Unfortunately, no one has determined any differences in the KLF4 or KLF5 levels in the younger population.

Comment 2. It’s mentioned that the group published a review on KLF4, KLF5, and KLF6 in colon cancer in 2008. How does this one differ? Are you only focusing on new evidence or expanding on what was published previously? If you are mentioning the previous review, then you should explain how this differs in the text.

Response 2. We provided a short statement to understand the difference between both reviews better.

Comment 3. For KLF4, loss seems to be the major effect in CRC, but there is one sentence that suggests KLF4 is important for cancer stem cells. This should be explained more regarding this contradiction.

Response 3. We included a paragraph describing KLF4's role as a stemness marker and discussed its potential mode of regulation in CRC stem cells.

Comment 4. Lines 107-113 seem to be largely repeats from earlier in the text and it’s unclear why talking about normal development here in the CRC section.

Response 4. We moved parts of this section to the section related to the KLF4 role in the normal intestinal epithelium and one sentence to the next paragraph. We updated the citations accordingly.

Comment 5. When talking about interferon gamma, the notation should be IFN-Y not INF-Y.

Response 5. We made appropriate modifications.

Comment 6. Lines 143-145 it is unclear why HDACis are being discussed. Is this in the presence of PPAR-y agonists? That is unclear in the text.

Response 6. We modified the text to improve its clarity.

Comment 7. For figures 2 and 3, more information should be included in the figure legend besides see text.

Response 7. We provided more information in both figures’ legends.

Comment 8. Does KLF5 regulate miRNAs? Is KLF5 regulated by miRNAs?

Response 8. We updated the section regarding miRs and KLF5 and added table 2, listing miRs regulating KLF5 expression.

Comment 9. Since KLF4 and KLF5 have such opposite effects in colon cancer, it might be beneficial to have a summarizing image or table to show the similarities/differences between the two.

Response 9. We provided an example of the opposite effects of KLF4 and KLF5 on the WNT signaling pathway in Figures 2 and 3 in the original manuscript. KLF4 is shown to work as a tumor suppressor, while KLF5 is an oncogene. KLF4 and KLF5 have been studied in the context of different pathways, and thus, it is rather challenging to provide a concise regulatory network.

Reviewer 2.

Comment 1. Besides colorectal cancers, what kind of cancers KLF4 and KLF5 could be involved in? Authors may briefly mention in in the introduction.

Response 1. In the introduction section, we listed several cancers involving both transcription factors and provided additional citations.

Comment 2. It would be nice to introduce KLF4 and KLF5 structures in the text too in more detail. I would also mention the significance of specific sites for ubiquitination, phosphorylation, acetylation, and sumoylation.

Response 2. We added a section in the introduction that addresses the posttranslational regulation of KLF4 and KLF5. In addition, we added two figures with protein alignment for KLF4 and KLF5 from Mus musculus and Homo sapiens and marked modified residues on these figures.

Comment 3. Were there any genetic mutations in KLF4 and KLF5 reported, involved in cancer, or in any diseases? If yes, which mutations were reported to impact cancer (or another disease?). A table may be needed for it.

Response 3. We accessed TCGA and COSMIC databases and identified mutations in KLF4 and KLF5 in CRC patients. These mutations are listed in Supplementary Tables 1 and 2. In addition, we added a short paragraph addressing some of the identified mutations.

Comment 4. A table may be needed on miRNAs, which may impact KLF5 regulation.

Response 4. We included Table 2, listing miRs regulating KLF5 expression.

Comment 5. Could the KLF4 and KLF5 mutations/dysfunction impact the radiotherapy response of patients?

Response 5. We provided information regarding the role of KLF4 and KLF5 in response to radiotherapy.

We revised the manuscript and addressed all the comments. We hope the incorporated changes will satisfy the Reviewers and Editor and render the revised manuscript suitable for publication. Thank you for being so considerate.

We confirm that neither the manuscript nor any parts of its content are currently under consideration or published in another journal.

All authors have approved the manuscript and agree with its submission to the Cancers.

Please feel free to contact me if I can be of further assistance,

Sincerely Yours,

Agnieszka B. Bialkowska, PhD

Associate Professor

Renaissance School of Medicine at Stony Brook University

Department of Medicine

GI Translational Research Lab

HSC-T17 Room 090

Stony Brook, NY 11794-8176

Phone: (631) 638 2161

Email: Agnieszka.Bialkowska@stonybrookmedicine.edu

Round 2

Reviewer 2 Report

The authors fulfilled my suggestions. Thank you.